# Synthesis of MOFs for RhB Adsorption from Wastewater

Qinhui Ren [1], Meng Nie [1], Lili Yang [1], Fuhua Wei [1,*], Bo Ding [1], Hongliang Chen [1], Zhengjun Liu [1] and Zhao Liang [2,*]

[1] College of Chemistry and Chemical Engineering, Anshun University, Anshun 561000, China; renqh@163.com (Q.R.); niemeng@126.com (M.N.); liliyang@163.com (L.Y.); ding99@yeah.net (B.D.); chenhledu@126.com (H.C.); liuzhengjun0427@163.com (Z.L.)

[2] State Key Laboratory of Advanced Design and Manufacturing for Vehicle Body, College of Mechanical and Vehicle Engineering, Hunan University, Changsha 410082, China

[*] Correspondence: yyspy@hnu.edu.cn (F.W.); walleliang@163.com (Z.L.)

**Abstract:** Fe-MOFs were prepared using a solvothermal method, and were characterized by scanning electron microscopy. We explored the application of Fe-MOFs as an adsorbing material for the removal of Rhodamine B (RhB) from aqueous solutions. The experimental data were simulated by dynamics and the results showed that the pseudo–second-order kinetics model was appropriate for analysis of RhB removal. We studied the adsorption capacity of MOF materials under different masses, concentrations, and pH conditions. When the pH was 6, the maximum adsorption capacity within 4 h was 135 mg/g. In summation, the removal of RhB from wastewater using MOFs is feasible, inexpensive, and effective. Hence, our findings indicate that MOFs have a broad application in the purification of wastewater.

**Keywords:** metal-organic frameworks; organic dye; wastewater

## 1. Introduction

The rapidly developing world economy, the continuously improving level of industrialization and the rapidly growing global population have given rise to a greater demand for water resource, whereas the increasingly serious water pollution has presented the world with an imminent water crisis. In particular, treatment of industrial wastewater has become an area of great concern. In recent decades, the chemical industry has undergone rapid development and the resulting environmental problems have affected the development of today's society. Industrial wastewater directly affects human life and health [1]. Dye wastewater is one of the main types of harmful industrial wastewater, and accounts for 10.1% of total wastewater [2]. Furthermore, printing and dyeing wastewater has become a serious threat to the environment, and due to its high content of organic pollutants, high alkalinity, effects on water quality and quantity, strong and variable color, toxicity, and complex composition, treating dye wastewater is difficult and has become a key issue in the safe processing of industrial wastewater. Moreover, common printing and dyeing wastewater pollution is not only toxic and deep in color, but also very difficult to degrade [3,4].

Dyes are ubiquitous and are widely used in plastics, carpets, printing, leather, food, and textiles. Due to their complex molecular structure, very low concentrations ($10^{-4} \sim 10^{-6}$ mol·L$^{-1}$) of dyes may pose a serious threat to ecosystems and human health [5]. Therefore, the treatment of dye wastewater has been an urgent problem in environmental fields. Rhodamine B (RhB) is a typical triphenylmethane dye, and is widely used in industrial fields such as cosmetic manufacturing, due to its bright and firm coloring [6]. However, in October 2017, the World Health Organization's International Agency for Research on Cancer (IARC) added RhB to the list of Category 3 carcinogens. The wastewater produced by the dye is carcinogenic, teratogenic, and mutagenic. If it is not effectively treated, but instead directly discharged into the external environment, it will have detrimental effects on the ecological environment as well as harm human health.

Traditional wastewater treatment methods include physical adsorption method, biological method, membrane separation method, chemical method, and the advanced oxidation method, among others. These methods have poor treatment effect [7] and are prone to secondary pollution. Sha et al. [8] combined silver iodide with UiO-66 to degrade RhB under visible light and obtained an enhanced photocatalytic effect. Zhao et al. [9] prepared AgI-mil-53 composite photocatalytic material by combining silver iodide and mil-53, which greatly enhanced the photocatalytic activity.

Since MOFs were reported in the 1990s, they have garnered much attention amongst different fields, such as energy storage [9,10], adsorption and separation [11–13], catalysis [14,15], drug delivery [16,17], carbon dioxide capture [18,19], chemical sensing [20], antibiotics [21,22], and others [23–27].

Fe is an element abundant in nature and has low toxicity. Therefore, it stands out as a metal element for MOFs. Because Fe-MOFs have many advantages over traditional porous materials, they can be used in gas storage, chemical separation and purification, catalysts, sensors, magnetic materials, optical devices, fluorescence and drug delivery. In this study, a high efficiency MOFs adsorbent was synthesized using the solvothermal method with $H_2BDC$ and $FeSO_4 \cdot 7H_2O$ as the main raw materials. The adsorbent has a good RhB removal effect.

## 2. Experimental Materials and Methods

### 2.1. Raw Materials

The terephthalic acid ($H_2BDC$, 98%),ferrous sulfate heptahydrate ($FeSO_4 \cdot 7H_2O$) ligands and organic dye RhB were purchased from Aladdin Biological Technology Co. Ltd.(Shanghai, China).

### 2.2. Preparation of Fe-MOFs

Briefly, 20mL Dimethylformamide (DMF) was added to a beaker with $H_2BDC$ (2.6734 g) and stirred for 30 min on a magnetic agitator. $FeSO_4 \cdot 7H_2O$ (3.2133 g) was dissolved in 20 mL of deionized water and the resulting aqueous solution was transferred to the DMF solution, stirred on a magnetic mixer for 30 min, and then transferred to the reactor to undergo a 12 h reaction at 150 °C. Finally, the reaction mixture was filtered and washed thoroughly with DMF and distilled water. The Fe-MOF was dried at 70 °C for 5 h.

### 2.3. Removal of RhB

The removal ability of Fe-MOFs was evaluated by using RhB as a model pollutant. Different concentrations (10, 20, 30, 40, and 50 ppm) of RhB solutions were prepared. The Fe-MOFs of different masses were added to each solution separately and stirred in natural light. The concentration was measured every 30 min or 60 min by UV-vis spectroscopy (554 nm) [28] to determine the adsorption capacity. The evaluation equation of adsorption quality and removal rate of Fe-MOFs on RhB is as follows:

$$q_e = \frac{(C_0 - C_e)V}{m} \tag{1}$$

$$\text{Removal rate\%} = (C_0 - C_t)/C_0 \times 100\% \tag{2}$$

where $C_0$, $C_t$, $C_e$, m, and V are the initial concentration of RhB, the concentration of RhB at t, the concentration of RhB at adsorption equilibrium, the mass of MOFs, and the volume of solution, respectively.

## 3. Results and Discussion

### 3.1. Preparation of Fe-MOFs

The Fe-MOFs were prepared by solvothermal synthesis.$H_2BDC$ and $FeSO_4 \cdot 7H_2O$ were dissolved in 20 mL DMF at a ratio of 1:1, stirred on a magnetic stirrer for 30 min, followed by in 12 h reaction in a 150 °C reaction kettle. The samples were then cooled

to room temperature, filtered, and the unreacted reactants were removed with DMF and deionized water, respectively. Unreacted organic chains were dissolved in organic solvents, and metal ions were dissolved in water. The remaining solids were new products, which were dried after washing.

### 3.2. Structural Characterization

XRD characterization of the materials is shown in Figure 1, which indicates that the material has no wide absorption peak, suggesting that the material has a good crystallinity.

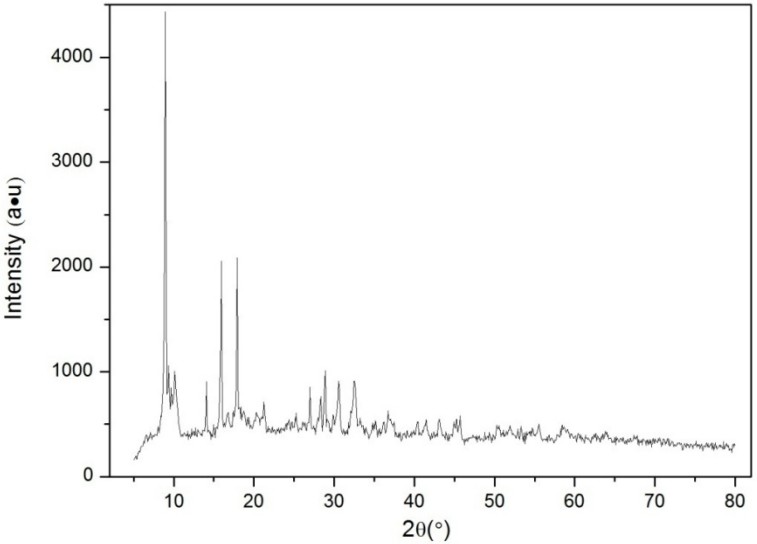

**Figure 1.** XRD of Fe-MOFs.

The BET surface area of Fe-MOFs was 21.4899 m$^2$/g, as shown in Figure 2. The average desorption pore diameter was 1.75 nm, and the adsorption average pore diameter was 8.6 nm, indicative of a mesoporous material.

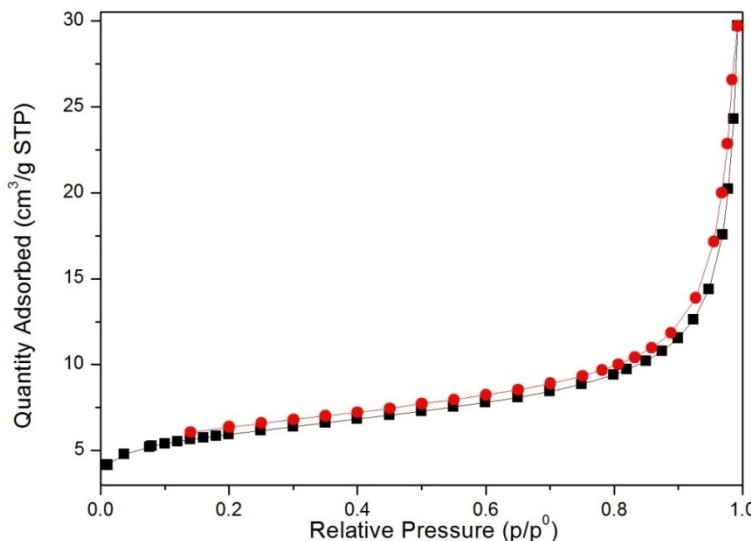

**Figure 2.** N$_2$ adsorption–desorption isotherms of Fe-MOFs.

Thermogravimetry (TG) curves of samples were recorded by a NETZSCH STA 449C thermal analyzer (Germany) in N$_2$ atmosphere in the range from 0 °C to 800 °C at a heating rate of 5 °C min$^{-1}$. According to the TG figure shown in Figure 3, the Fe-MOF material can be divided into three stages: (1) a mass loss (14.4%) between 20 °C and 300 °C, which

could be attributed to evaporation of residual solvents from the sample; (2) a mass loss of 40% between 301 °C and 509 °C, mainly due to the oxidation of the metal ion [29]; (3) a mass loss leaving a residual amount of 33.9% of the original mass, which begins at 610 °C, corresponding to the destruction of the framework structure. This result indicated that the material was stable below 410 °C. In most cases, MOFs generated via synthetic reactions between metal ions and organic ligands have a more stable structure [30].

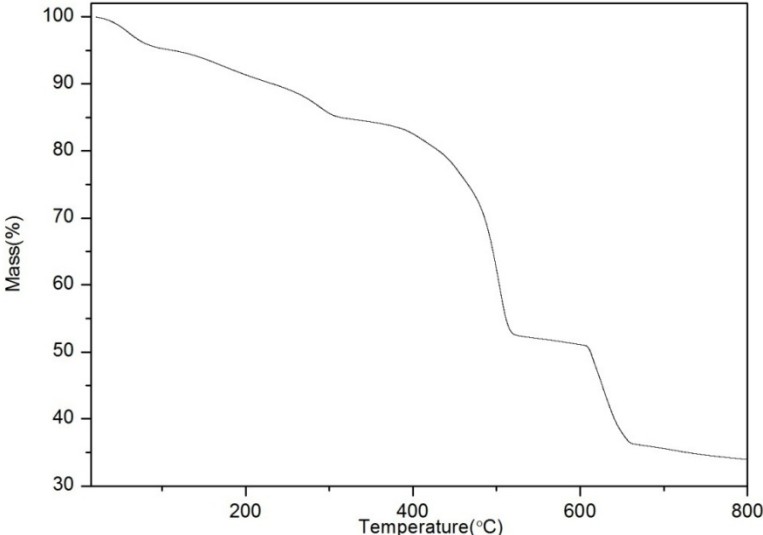

**Figure 3.** TG of Fe-MOFs.

The concentration of RhB in wastewater and the quality of MOFs were the main factors affecting the removal. When the concentration of RhB was 10, 20, 30, 40, and 50 ppm within 4h, the mass of the MOFs was 50 mg and the removal rate was 84.4, 28.4, 27, 19.3, and 14.9%, respectively, as shown in Figure 4. When the mass of Fe-MOFs was 20, 50, 100, and 200 mg, the concentration of RhB was 50 ppm and the removal rate was 20.25, 14.9, 27.88, and 87.03% within 4 h, respectively, increasing the concentration of RhB resulted in a gradual decrease in the removal rate. When the concentration of RhB was constant, the removal rate gradually increased with increasing mass of Fe-MOFs (Figure 4). The removal rate reached the maximum of 87% when the MOF was 200 mg. The color contrast of solution before and after removing RhB is shown in Figure 5. To test the reusability of Fe-MOFs, the used Fe-MOFs were filtered and added to a beaker filled with water, stirred on a magnetic mixer for 3 h, then filtered, dried, and reused. The result showed that the decline in the activity of Fe-MOFs was only 38% after four cycles, indicating a reasonable reusability.

To better understand Fe-MOF-mediated removal of RhB, the kinetics of RhB removal was tested. According to the literature, the removal of RhB is described by the following kinetic equation [31]—the pseudo–second-order (PSO) kinetic model:

$$\frac{t}{q_t} = \frac{t}{q_e} + \frac{1}{k_2 q_e^2} \tag{3}$$

where t is the reaction time (min), $k_2$ is the reaction rate constant (g(mg min)$^{-1}$)), and $q_t$ and $q_e$ represent the adsorption capacity at t and equilibrium, respectively. The adsorption results of Fe-MOFs on RhB are shown in Figure 6, Tables 1 and 2. The pseudo–second-order model appropriately describes the removal of RhB.

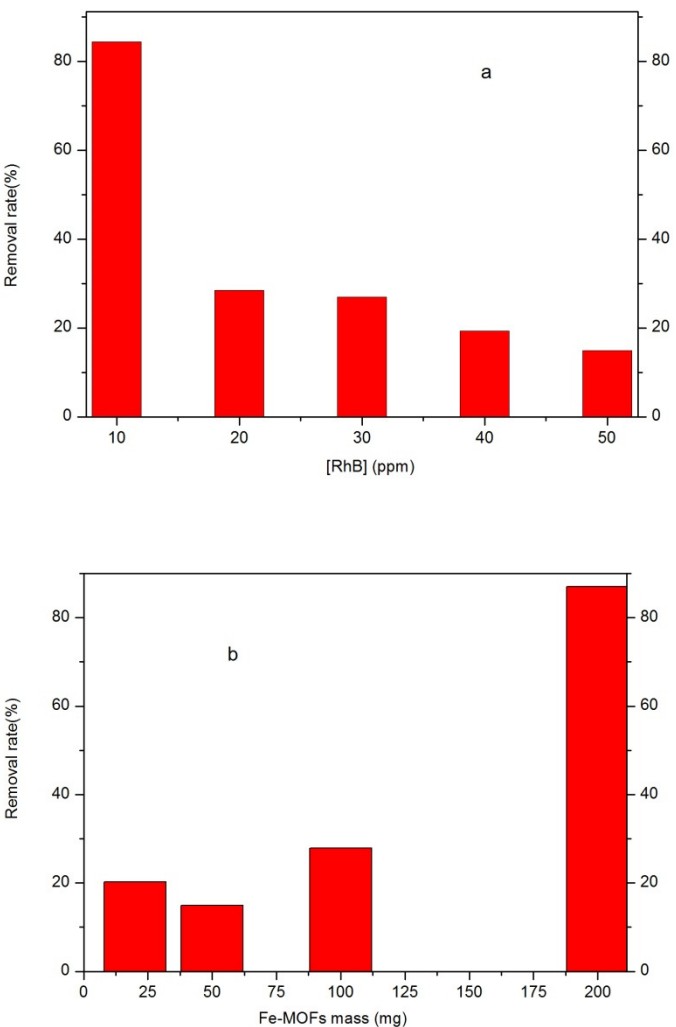

**Figure 4.** The removal rate of Fe-MOFs on RhB (**a**) 50 mg of Fe-MOFs (**b**) 50 ppm of RhB.

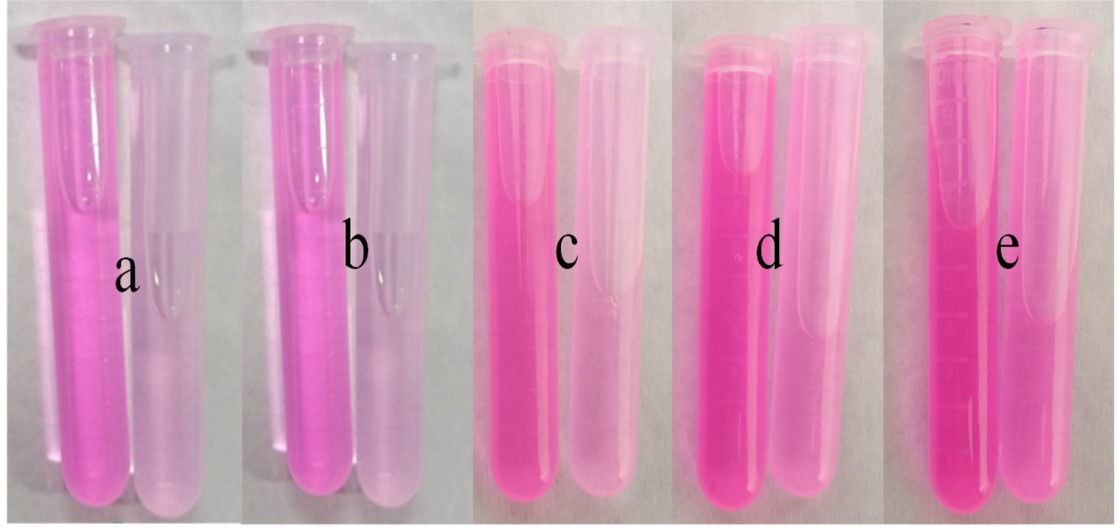

**Figure 5.** Color comparison of solution before and after RhB removal (**a**) 10 mg/L; (**b**) 20 mg/L; (**c**) 30 mg/L; (**d**) 40 mg/L; (**e**) 50 mg/L.

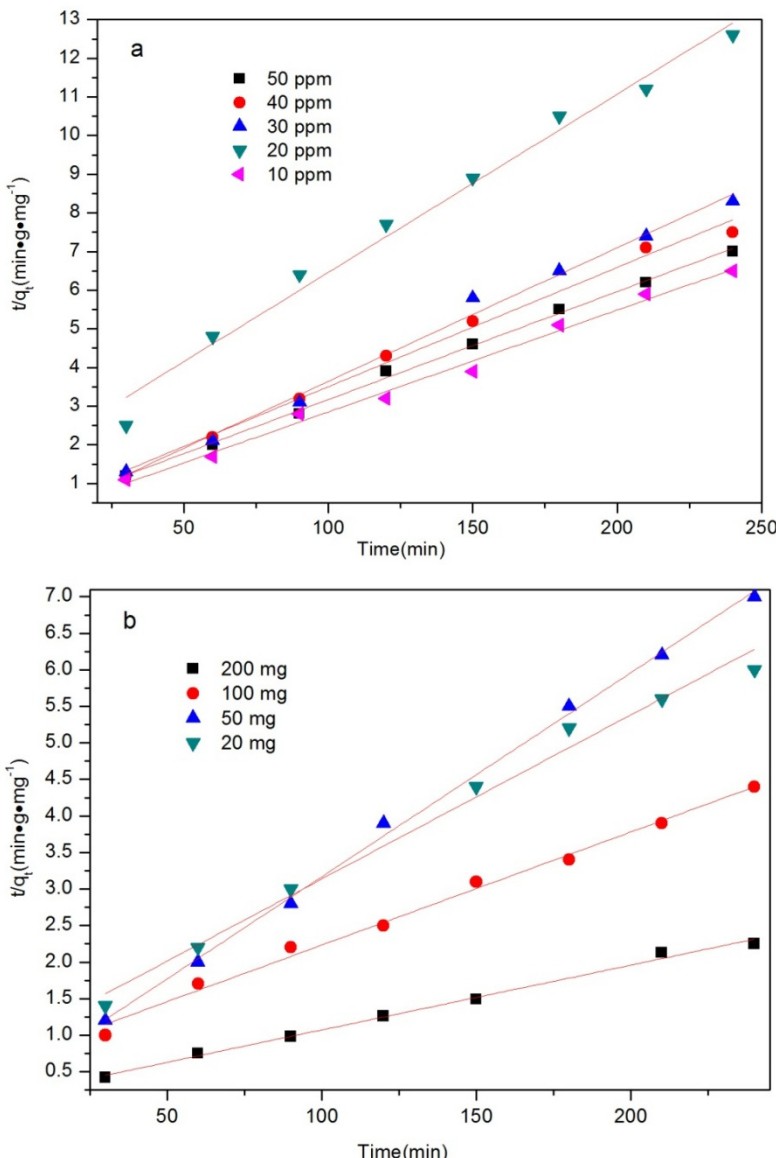

**Figure 6.** Pseudo–second-order kinetic model of Fe-MOFs on RhB (**a**) 50 mg, 20 °C; (**b**) 50 ppm, 20 °C.

**Table 1.** The PSO parameters of Fe-MOFs with different concentrations of RhB.

| Concentration (ppm) | K (g(mg min)$^{-1}$) | R$^2$ |
|---|---|---|
| 50 | $2.8 \times 10^{-2}$ | 0.9976 |
| 40 | $3.1 \times 10^{-2}$ | 0.992 |
| 30 | $3.5 \times 10^{-2}$ | 0.992 |
| 20 | $4.6 \times 10^{-2}$ | 0.983 |
| 10 | $2.6 \times 10^{-2}$ | 0.9906 |

**Table 2.** The PSO parameters of RhB with Fe-MOFs of different masses.

| Mass (mg) | R$^2$ | K (g(mg min)$^{-1}$) |
|---|---|---|
| 200 | 0.9941 | $8.8 \times 10^{-3}$ |
| 100 | 0.9921 | $1.5 \times 10^{-2}$ |
| 50 | 0.9976 | $2.8 \times 10^{-2}$ |
| 20 | 0.987 | $2.2 \times 10^{-2}$ |

The adsorption of RhB by MOF can be partly attributed to $\pi$–$\pi$ interactions and hydrogen bonding [32–34]. Additionally, since the amine group of the RhB is basic, the chemisorption mechanism relies heavily on acid-base interactions. Based on this knowledge, we tested the effect of pH on the adsorption performance. For more details, refer to the Experimental section. The results are shown in Figure 7, and indicated that pH greatly affected the adsorption performance. The adsorption capacity initially increased with increasing pH, peaking at pH = 6 before decreasing as the pH of the solution continued to rise. This phenomenon can be attributed to the presence of electric charges of RhB molecules as well as the electrostatic interactions that dominate in the adsorption process [35].

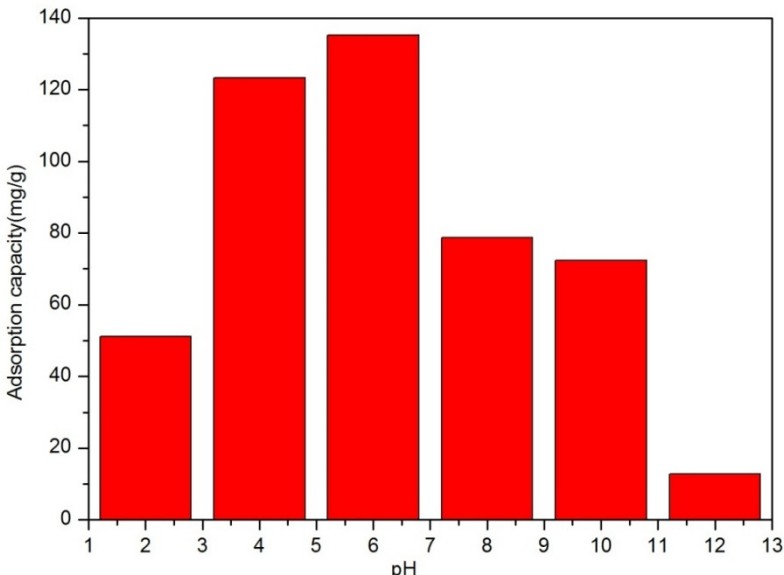

**Figure 7.** Adsorption Capacity of different pH on Fe-MOFs (50 mg, 50 ppm, 20 °C).

At a certain temperature, Langmuir isotherms and Freundlichisotherms were used to explore the relationship between the adsorption capacity of the MOF materials and the equilibrium concentration of RhB. According to the adsorption parameters, the following equations were used to determine the efficiency of the adsorbent to remove RhB from the environment:

$$C_e/q_e = C_e/q_{max} + 1/k_L q_{max} \tag{4}$$

$$\ln q_e = \frac{1}{n}\ln C_e + \ln k_f \tag{5}$$

where $C_e$ (mg/L) is the equilibrium concentration of CR solution, $q_e$ (mg/g) is the equilibrium adsorption amount, $q_{max}$ (mg/g) is the Langmuir constant (maximum adsorption capacity) that can be obtained from the reciprocal of the slope of a plot of $C_e/q_e$ versus $C_e$, b (L/mg or L/mol) is the Langmuir constant related to the free energy of adsorption, and $K_f$ (L/mg) is the Freundlich constant.

As shown in Figures 8 and 9 and Table 3, the removal of Congo red by Fe-MOFs was consistent with the two adsorption models, which can well explain the removal efficiency of environmental pollutants using Fe-MOFs.

**Table 3.** Summary of Langmuir and Freundlich isotherm constants for the removal of RhB by Fe-MOFs.

| Adsorbent | Langmuir Isotherm | | Freundlich Isotherm | |
|---|---|---|---|---|
| | K(L/mg) | $R^2$ | K (mg/g)(L/mg)$^{1/n}$ | $R^2$ |
| Fe-MOFs | 0.0131 | 0.99783 | 0.78938 | 0.99031 |

In summary, the main mechanism of adsorption of RhB is as follows, (1) RhB and MOFs contain benzene rings, which can be adsorbed together by π–π action; (2) MOFs are porous materials, and part of RhB is adsorbed through the void; (3) the molecules contain oxygen and nitrogen atoms, which can be adsorbed by hydrogen bonds; (4) the molecules can be adsorbed together by electrostatic action.

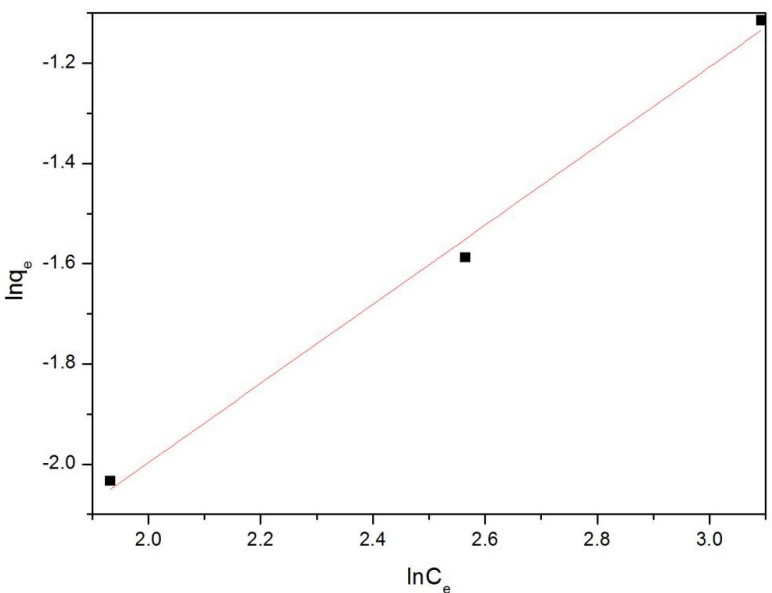

**Figure 8.** Freundlich isotherm for the adsorption of CR on Fe-MOFs (50 mg, 50 ppm, 20 °C).

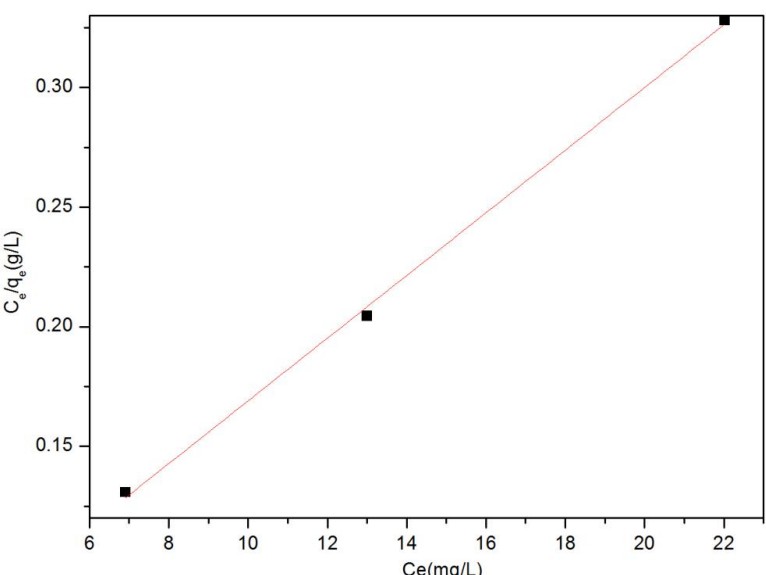

**Figure 9.** Langmuir isotherm for the adsorption of CR on Fe-MOFs (50 mg, 50 ppm, 20 °C).

## 4. Conclusions

In this study, we successfully prepared Fe-MOFs using a solvothermal method, which were used to remove RhB. Different concentrations of RhB and different masses of Fe-MOFs were used, and the removal of RhB by Fe-MOFs yielded good results. The experimental results accorded with the pseudo–second-order kinetic model, with a good correlation coefficient, demonstrating that the theory and practice were in good agreement. We also examined the influence of solution pH on adsorption. Our findings showed that at pH 6, the adsorption effect of Fe-MOFs on RhB was the best, reaching 136 mg/g. We also

investigated the adsorption mechanism of RhB by Fe-MOFs. Taken together, our results suggest that Fe-MOFs could effectively remove RhB, and therefore have good prospects for practical application.

**Author Contributions:** Conceptualization, F.W. and M.N.; methodology, L.Y.; software, B.D.; validation, Q.R., H.C. and F.W.; formal analysis, F.W.; investigation, M.N.; resources, Q.R.; data curation, Z.L. (Zhao Liang); writing—original draft preparation, Q.R.; writing—review and editing, F.W.; visualization, Z.L. (Zhengjun Liu). All authors have read and agreed to the published version of the manuscript.

**Funding:** Guizhou Education Department Youth Science and Technology Talents Growth Project (KY[2019]149) and (KY[2020]132).

**Institutional Review Board Statement:** Not applicable.

**Informed Consent Statement:** Not applicable.

**Data Availability Statement:** The data presented in this study are available on request from the corresponding author.

**Acknowledgments:** This work was supported by Guizhou Education Department Youth Science and Technology Talents Growth Project (KY[2019]149) and (KY[2020]132).

**Conflicts of Interest:** The authors declare no conflict of interest.

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
