# Peer review of "Synthesis of MOFs for RhB Adsorption from Wastewater"

_inorganics, doi:10.3390/inorganics10030027_

Round 1
Reviewer 1 Report
The authors describe the synthesis of MOFs for RhB adsorption from wastewater. The idea is really interesting and the used methods are appropriate for such kind of research, however, the manuscript contains a few but crucial drawbacks. The most important is that it is written with many inexactnesses, i.g.:
- All the calculations are done without statistical analysis: there are no errors, and significant digits are used in excess, see Tables 1 and 2.
- Many sentences are unclear: "second dynamic model" (line 204) - which one?; "The equation is as follows" (eq.1) - what equation? what does it describe?)
- Figs. 3 and 4 are of poor quality: it's hard to say what mass should be assigned to the bar on the right in Fig. 4, for example. Additionally, the OY axis is not the "Removal rate" but it is a percent of remaining dye in the solution and it is inversely proportionate to the removal efficiency! Note, that now the "removal rate" is slower when concentrations of reactants are higher!
- The problem of the "Removal rate" is visible also in lines 125-127 (higher reactant conc. reduces the reaction rate...).
- Discussion on thermal analysis is also not correct, in my opinion. The product is clearly stable below 400°C, not 610°C (Fig. 3), additionally, oxidation of iron should increase the sample mass (lines 112-113).
Minor problems:
- Discussion on the pH dependence is too general (lines 166-168). There is no information about pKa of RhB, for example.
- Explanation of equations, e.g. eq. (4) and kF is not completed.
Taking into account the listed problems and the fact that it is the manuscript after the first revision, I suggest a major revision of the manuscript in the present form because it still needs an important improvement.
Author Response
Dear Reviewers:
Thank you for your letter and for the reviewers’ comments concerning our manuscript entitled “Synthesis of MOFs for RhB adsorption from wastewater(inorganics-1585932)”. Those comments are all valuable and very helpful for revising and improving our paper. We have studied comments carefully and tried our best to revise the manuscript. Revised portion are marked in red in the paper. The point to point responds to the reviewer’s comments are listed as following:
Responds to the reviewer’s comments:
Reviewer#1
The authors describe the synthesis of MOFs for RhB adsorption from wastewater. The idea is really interesting and the used methods are appropriate for such kind of research, however, the manuscript contains a few but crucial drawbacks. The most important is that it is written with many inexactnesses, i.g:
Comment:1. All the calculations are done without statistical analysis: there are no errors, and significant digits are used in excess, see Tables 1 and 2.
Response to Comment: Thank you for your careful reading of our manuscript. We are sorry about the mistakes. According to your comment, we have corrected these comment in the revised paper. Revised portion are marked in red in the paper. (Page 8,Tab.1 and 2.)
Comment:2. Many sentences are unclear: "second dynamic model" (line 204) - which one?; "The equation is as follows" (eq.1) - what equation? what does it describe?).
Response to Comment: Thank you for your careful reading of our manuscript. We are sorry about the mistakes.According to your comment, the new manuscript has been thoroughly edited by our native English speaking editors.
Comment:3. Figs. 3 and 4 are of poor quality: it's hard to say what mass should be assigned to the bar on the right in Fig. 4, for example. Additionally, the OY axis is not the "Removal rate" but it is a percent of remaining dye in the solution and it is inversely proportionate to the removal efficiency! Note, that now the "removal rate" is slower when concentrations of reactants are higher!
Response to Comment: Thank you for your careful reading of our manuscript. We are sorry about the mistakes. According to your comment, we have corrected these mistakes in the new paper. Figs. 3 and 4 have replaced by new pictures in the revised paper.
Comment: 4.The problem of the "Removal rate" is visible also in lines 125-127 (higher reactant conc. reduces the reaction rate...).
Response to Comment:Thank you for your valuable suggestion. According to your comment, We have amended the content in the revised paper.Revised portion are marked in red in the paper. (Page 5.).
Comment:5.Discussion on thermal analysis is also not correct, in my opinion. The product is clearly stable below 400°C, not 610°C (Fig. 3), additionally, oxidation of iron should increase the sample mass (lines 112-113).
Response to Comment: Thank you for your careful reading of our manuscript. We are sorry about the mistakes. According to your comment, we have corrected the data in the revised paper. Revised portion are marked in red in the new paper.Revised portion are marked in red in the paper. (Page 4.).
Comment:6.Discussion on the pH dependence is too general (lines 166-168). There is no information about pKa of RhB, for example.
Response to Comment: Thank you for your valuable advice. We only adjusted the pH of RhB solution, and did not think of testing the pH of RhB solution.It will be further studied in future studies.
Comment:7.Explanation of equations, e.g. eq. (4) and kF is not completed.
Response to Comment: Thank you for your valuable suggestion. According to your comment, We have added kf in the revised paper. Revised portion are marked in red in the paper. (Page 9.)

Reviewer 2 Report
The authors improved their manuscript based on reviewers' comments. Hence, I can recommend it for acceptance.
Author Response
Thank you for your review.

Round 2
Reviewer 1 Report
Dear Authors,
I have read your answers carefully. After that, I have listed my comment below and suggested major revision again:
- Comment 1. Errors in Tab. 1 and 2: Unfortunately, I do not see errors (e.g. 35 ± 3). Probably, we thought about something different... It is true that the Tables look much better but still there are no errors.
- Comment 3. Sorry, but I see the same figures in the 'v2' of the manuscript, so there is no change.
- Comment 4. There is no 'portion' word in the manuscript. I still see the 'removal rate' - it is not corrected in 'v2'.
- Comment 7. It is done, OK. However, there is Kf in the text but kf in the equation (4).
Generally: errors, the correct definition of the 'rate', and corrected figures ('removal rate' is also in Figs + e.g. 'Concentration (ppm)' (conc. of what?)) are crucial here.
I can say that other problems are solved now.
Author Response
Dear Reviewers:
Thank you for your letter and for the reviewers’ comments concerning our manuscript entitled “Synthesis of MOFs for RhB adsorption from wastewater(inorganics-1585932)”. Those comments are all valuable and very helpful for revising and improving our paper. We have studied comments carefully and tried our best to revise the manuscript. Revised portion are marked in red in the paper. The point to point responds to the reviewer’s comments are listed as following:
Responds to the reviewer’s comments:
Reviewer#1
I have read your answers carefully. After that, I have listed my comment below and suggested major revision again:
Comment:1. Comment 1. Errors in Tab. 1 and 2: Unfortunately, I do not see errors (e.g. 35 ± 3). Probably, we thought about something different... It is true that the Tables look much better but still there are no errors.
(Comment:1 All the calculations are done without statistical analysis: there are no errors, and significant digits are used in excess, see Tables 1 and 2.)
Response to Comment: Thank you for your careful reading of our manuscript. The RhB concentration of preparation is not necessarily accurate, but the data is subject to UV spectrometer test in the manuscript, the actual value error of concentration is less than 5%; The mass of MOFs is accurately weighed with an analytical balance to within one thousandth of an error, and K and R2 are simulated by Oringin.
Comment:2.Comment 3. Sorry, but I see the same figures in the 'v2' of the manuscript, so there is no change.(Comment:3. Figs. 3 and 4 are of poor quality: it's hard to say what mass should be assigned to the bar on the right in Fig. 4, for example. Additionally, the OY axis is not the "Removal rate" but it is a percent of remaining dye in the solution and it is inversely proportionate to the removal efficiency! Note, that now the "removal rate" is slower when concentrations of reactants are higher!)
Response to Comment: Thank you for your careful reading of our manuscript. The data of OY axis was calculated by formula(removal rate%=(1-Ct /C0) x100%), so there was no change in the data of Fig 4 in the "V2"of the manuscript.
Comment: 3. Comment 4. There is no 'portion' word in the manuscript. I still see the 'removal rate' - it is not corrected in 'v2'.(Comment: 4.The problem of the "Removal rate" is visible also in lines 125-127 (higher reactant conc. reduces the reaction rate...).
Response to Comment: Thank you for your review. We are sorry about the question. The data of removal rate was calculated by formula(removal rate%=(1-Ct /C0) x100%).We may not understand you yet, we hope you will be more careful.
Comment: 4.Comment 7. It is done, OK. However, there is Kf in the text but kf in the equation (4)(Comment:7.Explanation of equations, e.g. eq. (4) and kF is not completed.).
Response to Comment: Thank you for your review.
Maybe I didn't understand what you meant, so I couldn't explain some problems in detail. I hope I can get detailed guidance and pay more attention to it in future research.

Round 3
Reviewer 1 Report
Dear Authors,
Thank you for your quick answers. I would like to make my suggestions more clear:
- Statistical analysis is indispensable! You can not fill up tables with calculated data without estimation of their errors. We can not say anything about results whose accuracy is unknown. So, you have to estimate errors for 'K' (Tables 1, 2, and 3) and 'k' (Table 3). It is also a good idea to describe the procedure used for that (but it is not necessary). You calculate values of K and k based on some other parameters. If even one of them is an experimental result with even a small error, this error influences the error of the final result ('K' or 'k') ("propagation of uncertainty"). Then, using derivatives, you can estimate errors for your 'K' and 'k' parameters. In my opinion, there is no possibility to publish a manuscript where calculated parameters are presented without their errors. It is not a solution, to use more significant digits. The number of significant digits should be limited by the error order.
- In my opinion the term "removal rate" is incorrect. "Rate" is a word with a strict connotation to kinetics and describes changes of e.g. concentration in time (rate = constant * concentration(s)).
The "removal rate" (r.r.) in the form (1-Ct /C0) x100% is something different. It shows a level of a reactant concentration that is present in the mixture after a particular time (what time? - this information should be added). When we calculate r.r. at different moments of the adsorption process, the r.r. can decrease from 100% to 0%, so, the r.r. is a measure of the reactant (RhB) concentration. If it is true, the lower RhB concentration after a particular time used for calculation (not presented in the manuscript) (= lower r.r.) indicates the higher degree of adsorption (higher "rate of the reaction")!
Let's analyze Fig. 4a. Higher concentration of RhB (reactant) (it should be written on the axis in the form "[RhB] (ppm)", not "Concentration(ppm)") leads to lower r.r. but it means that the degree of adsorption is higher! If you increase [RhB] in the mixture, its adsorption is higher because it is easier for Fe-MOF to meet RhB molecules. This is why I have written that the "removal rate" is not the rate but it is a percent of dye remaining in the solution and it is inversely proportionate to the removal efficiency and a true removal rate!
What about Fig. 4(b)? I do not understand why r.r. is so high for 200 mg of Fe-MOF (it should be noted on the axis: "Fe-MOF mass (mg)" instead of a mysterious "Mass(mg)"). If you introduce, let's say, 25 mg of Fe-MOFs to the reaction mixture and after the particular time (not presented in the manuscript) there is only ca. 20% of starting concentration of RhB (what you call "removal rate", in the manuscript and Fig. 4), the addition of more Fe-MOFs should increase the process because effective collisions between Fe-MOFs and RhB molecules are more likely OR there should be no change in the adsorption rate (and next decrease of RhB percentage below 20%) because the system reached the maximum rate of adsorption and additional Fe-MOF surface changes nothing. It needs comment, of course, but it looks strange.
Shortly speaking: the "removal rate" is not the rate but the percentage of the RhB remaining in the reaction mixture after the time where the calculation was done (this time (and (1-Ct /C0) x100% ?) should be presented in the manuscript). Thus, the higher the "removal rate" defined in this way, the slower the adsorption process! It has to be solved, definitely! - Description of Fig. 4 is not complete:
(*) I do not know what is 50 mg and 50 ppm. You can write "a. 50 mg of Fe-MOFs; b. 50 ppm of RhB", for example.
(*) The bars presented in Fig. 4(b) are too wide - it looks bad.
(*) The equation defining r.r. (after the change of this incorrect name) can be presented even in the description of Fig. 4 or instead of "removal rate" in that label of the OY axis.
In summary, in my opinion, the key problems are error of K and k, and correction of the "removal rate" term.
Additionally, the quality of Fig. 4 and its description should be improved.
Author Response
Dear reviewer:
Thank you for your letter and for the reviewers’ comments concerning our manuscript entitled “Synthesis of MOFs for RhB adsorption from wastewater(inorganics-1585932)”. Those comments are all valuable and very helpful for revising and improving our paper. We have studied comments carefully and tried our best to revise the manuscript. Revised portion are marked in red in the paper. The point to point responds to the reviewer’s comments are listed as following:
Responds to the reviewer’s comments:
Comment: 1.Statistical analysis is indispensable! You can not fill up tables with calculated data without estimation of their errors. We can not say anything about results whose accuracy is unknown. So, you have to estimate errors for 'K' (Tables 1, 2, and 3) and 'k' (Table 3). It is also a good idea to describe the procedure used for that (but it is not necessary). You calculate values of K and k based on some other parameters. If even one of them is an experimental result with even a small error, this error influences the error of the final result ('K' or 'k') ("propagation of uncertainty"). Then, using derivatives, you can estimate errors for your 'K' and 'k' parameters. In my opinion, there is no possibility to publish a manuscript where calculated parameters are presented without their errors. It is not a solution, to use more significant digits. The number of significant digits should be limited by the error order.
Response to Comment:Thank you for your careful reading of our manuscript. According to your comment, we have corrected these comment in the revised paper. Revised portion are marked in red in the paper. (Page 8,Tab.1 and 2;page 13,Tab.3.)
Comment: 2.In my opinion the term "removal rate" is incorrect. "Rate" is a word with a strict connotation to kinetics and describes changes of e.g. concentration in time (rate = constant * concentration(s)).
The "removal rate" (r.r.) in the form (1-Ct /C0) x100% is something different. It shows a level of a reactant concentration that is present in the mixture after a particular time (what time? - this information should be added). When we calculate r.r. at different moments of the adsorption process, the r.r. can decrease from 100% to 0%, so, the r.r. is a measure of the reactant (RhB) concentration. If it is true, the lower RhB concentration after a particular time used for calculation (not presented in the manuscript) (= lower r.r.) indicates the higher degree of adsorption (higher "rate of the reaction")!
Let's analyze Fig. 4a. Higher concentration of RhB (reactant) (it should be written on the axis in the form "[RhB] (ppm)", not "Concentration(ppm)") leads to lower r.r. but it means that the degree of adsorption is higher! If you increase [RhB] in the mixture, its adsorption is higher because it is easier for Fe-MOF to meet RhB molecules. This is why I have written that the "removal rate" is not the rate but it is a percent of dye remaining in the solution and it is inversely proportionate to the removal efficiency and a true removal rate!
What about Fig. 4(b)? I do not understand why r.r. is so high for 200 mg of Fe-MOF (it should be noted on the axis: "Fe-MOF mass (mg)" instead of a mysterious "Mass(mg)"). If you introduce, let's say, 25 mg of Fe-MOFs to the reaction mixture and after the particular time (not presented in the manuscript) there is only ca. 20% of starting concentration of RhB (what you call "removal rate", in the manuscript and Fig. 4), the addition of more Fe-MOFs should increase the process because effective collisions between Fe-MOFs and RhB molecules are more likely OR there should be no change in the adsorption rate (and next decrease of RhB percentage below 20%) because the system reached the maximum rate of adsorption and additional Fe-MOF surface changes nothing. It needs comment, of course, but it looks strange.
Shortly speaking: the "removal rate" is not the rate but the percentage of the RhB remaining in the reaction mixture after the time where the calculation was done (this time (and (1-Ct /C0) x100% ?) should be presented in the manuscript). Thus, the higher the "removal rate" defined in this way, the slower the adsorption process! It has to be solved, definitely!
Response to Comment: Thank you for your review. According to your comment, we have corrected these comment in the revised paper. Revised portion are marked in red in the paper. (page 2,page 5 and page 6,Fig 4)
Comment: 3.Description of Fig. 4 is not complete:
(*) I do not know what is 50 mg and 50 ppm. You can write "a. 50 mg of Fe-MOFs; b. 50 ppm of RhB", for example.
(*) The bars presented in Fig. 4(b) are too wide - it looks bad.
(*) The equation defining r.r. (after the change of this incorrect name) can be presented even in the description of Fig. 4 or instead of "removal rate" in that label of the OY axis.
Response to Comment: Thank you for your careful reading of our manuscript. According to your comment, we have corrected these comment in the revised paper. (Page 6, Fig. 4.)
Comment: 4.In summary, in my opinion, the key problems are error of K and k, and correction of the "removal rate" term.Additionally, the quality of Fig. 4 and its description should be improved.
Response to Comment: Thank you for your review. According to your comment, we have corrected these comment in the revised paper. (Page 6, Fig. 4.)
